# Importance of Mitochondria in Cardiac Pathologies: Focus on Uncoupling Proteins and Monoamine Oxidases

**DOI:** 10.3390/ijms24076459

**Published:** 2023-03-30

**Authors:** Rainer Schulz, Klaus-Dieter Schlüter

**Affiliations:** Institute of Physiology, Justus-Liebig University, 35392 Giessen, Germany

**Keywords:** ischemia, reperfusion, heart failure, pulmonary hypertension, reactive oxygen species

## Abstract

On the one hand, reactive oxygen species (ROS) are involved in the onset and progression of a wide array of diseases. On the other hand, these are a part of signaling pathways related to cell metabolism, growth and survival. While ROS are produced at various cellular sites, in cardiomyocytes the largest amount of ROS is generated by mitochondria. Apart from the electron transport chain and various other proteins, uncoupling protein (UCP) and monoamine oxidases (MAO) have been proposed to modify mitochondrial ROS formation. Here, we review the recent information on UCP and MAO in cardiac injuries induced by ischemia-reperfusion (I/R) as well as protection from I/R and heart failure secondary to I/R injury or pressure overload. The current data in the literature suggest that I/R will preferentially upregulate UCP2 in cardiac tissue but not UCP3. Studies addressing the consequences of such induction are currently inconclusive because the precise function of UCP2 in cardiac tissue is not well understood, and tissue- and species-specific aspects complicate the situation. In general, UCP2 may reduce oxidative stress by mild uncoupling and both UCP2 and UCP3 affect substrate utilization in cardiac tissue, thereby modifying post-ischemic remodeling. MAOs are important for the physiological regulation of substrate concentrations. Upon increased expression and or activity of MAOs, however, the increased production of ROS and reactive aldehydes contribute to cardiac alterations such as hypertrophy, inflammation, irreversible cardiomyocyte injury, and failure.

## 1. Introduction

An imbalance in the formation and removal of reactive oxygen species (ROS) leads to oxidative stress, which plays a role in the development of cardiovascular diseases, such as hypertension [1], hypertrophy [2], myocardial injury following ischemia and reperfusion (I/R), and heart failure [3,4,5]. However, ROS are also involved in numerous signaling pathways related to cell metabolism, growth and survival [4,5]. ROS are molecules that have one or more unpaired electrons (i.e., superoxide and hydroxyl) or are non-radicals which can generate free radicals (i.e., hydrogen peroxide). Intracellular ROS are derived from various enzymes (xanthine oxidase, uncoupled nitric oxide synthase, sodium-potassium ATPase, and nicotinamide adenine dinucleotide phosphate (NADPH) oxidase). Moreover, in mitochondria apart from the electron transport chain a number of proteins have been identified which contribute to ROS formation, namely connexin 43 [6], signal transducer and activator of transcription [7], or p66shc [8], as previously discussed by us (for review, also see [9]).

We now concentrate on two other mitochondrial proteins, i.e., uncoupling proteins (UCP, depending membrane on the mitochondrial membrane potential) and monoamine oxidase (MAO). The present review will summarize recent findings about UCP and MAO connected to reperfusion injury, cardiac protection, and heart failure secondary to I/R injury or pressure overload.

## 2. Uncoupling proteins (UCP)

### 2.1. Mitochondrial Reactive Oxygen Species and Cardiac Ischemia/Reperfusion Injury

Mitochondria are at the crossroads of cell death and survival through a plethora of functions [10]. During ischemia, the interruption in the generation of mitochondrial adenosine triphosphate by oxidative phosphorylation is the triggering mechanism for the profound ionic and biochemical disturbances of cardiomyocytes (see above), the duration of which determines the fate of the cells [11]. Upon reperfusion, the abnormal resumption of mitochondrial respiration (leading to an excessive and unregulated ROS production) and mitochondrial matrix calcium accumulation can synergistically exacerbate energy collapse through the opening of the permeability transition pore, a pathological disruption of the inner membrane that induces massive matrix swelling and culminates in cell death [12]. In support of this, mice in which mitochondrial calcium overload was inhibited by cardiomyocyte-specific deletion of the mitochondrial calcium uniporter were protected against I/R injury [13].

The attenuation of mitochondrial ROS generation by different interventions, including either pharmacological or genetic inhibition of the reverse electron transport from complex II to complex I limits I/R injury in rodents [14,15,16,17,18] and in the in vivo pig model [19,20]. Furthermore, STAT 3 activation occurs during I/R and impacts mitochondrial function by decreasing ROS formation in rat and mouse mitochondria [21] through the inhibition of complex I [22].

Different subpopulations of cardiac mitochondria are known, namely subsarcolemmal, interfibrillar and perinuclear mitochondria, which undergo fusion and fission processes. Decreasing mitochondrial fission by genetic or pharmacological inhibition of the fission protein dynamin-related protein 1 mitigated cardiac injury in murine models of I/R [23]. Such an approach, however, failed to protect the heart in the more clinically relevant closed-chest pig model of acute myocardial infarction [24].

As for their subcellular location, subsarcolemmal mitochondria have a greater contribution to ROS production [25] and I/R injury [26] than interfibrillar mitochondria. Moreover, only subsarcolemmal mitochondria contain connexin 43 at their inner membrane [27], a protein involved in mitochondrial ROS formation [6,28,29].

### 2.2. Metabolism and Ischemia/Reperfusion Injury

It has long been known that cardiac substrate metabolism is a main determinant of the severity of I/R injury. One of the first cardioprotective strategies examined against I/R injury was a metabolic treatment, employing glucose–insulin–potassium infusions to attenuate electrographic disturbances during myocardial infarction [30]. It is now known that almost every specific metabolic substrate pathway can affect cardiac I/R injury [31]. In terms of metabolic approaches, activation of glycolysis, glucose and ketone oxidation, and inhibition of fatty acid metabolism and oxygen consumption hold the most promise for protecting the heart against I/R injury [31]. Increases in glucose uptake and glycolysis in rodent hearts are mandatory for protecting the heart during low-flow ischemia [32]. Increased fatty acid uptake and incomplete fatty acid metabolism during ischemia aggravate I/R injury through increased mitochondrial ROS production [33] and loss of endogenous cardioprotection [34]. In support of this finding, Oeing et al. [35] recently showed that activated glycolysis at the expense of fatty acid oxidation offers protection against I/R injury in the murine heart.

## 3. Uncoupling Proteins

UCPs are proteins that are located at the inner mitochondrial membrane and have potential protonophore functions. A protonophore reduces the proton gradient across the inner mitochondrial membrane, the driving force of the mitochondrial ATP synthase, also known as complex V. The consequence is the uncoupling of electron transport from ATP generation, producing heat instead of ATP. This basic function of UCPs is highly conserved during evolutionary processes. I.e., cold-induced induction of UCP expression in *Lacerta vivipara* produces heat to avoid freezing at sub-zero temperatures [36]. At the same time, it allows the preservation of glucose, which has cryoprotective properties. Therefore, UCPs have two different functions, heat production and inhibiting the oxidative consumption of glucose [37]. Both functions of UCPs can also be seen in higher vertebrates such as mammalians. However, the exact role of this pathway in homeothermic animals is less clear.

In line with these observations, fatty acids induce the expression of UCPs. It has been proposed that UCPs transport fatty acids out of the matrix of mitochondria to avoid lipotoxicity. The proton gradient drives fatty acid export. Therefore, a high expression of UCP induced by fatty acids lowers the proton gradient across the inner mitochondrial membrane, lowers mitochondrial membrane potential, and thereby increases the use of fatty acids. At the same time, increasing the use of fatty acids as fuel will inhibit glucose oxidation. Therefore, UCPs may act as metabolic sensors that participate in directing fuel consumption from glucose to fatty acids. Consequently, the downregulation of UCP2 should favor the use of glucose. These suggestions have previously been reviewed in detail [38]. In this review, we will focus on the consequences of UCP expression in tissues in terms of I/R.

High mitochondrial membrane potentials inhibit the electron transport velocity of the electron transport chain. Under such conditions, electrons are transferred to oxygen, leading to the formation of superoxide radicals. Mitochondria have an electron defense system that allows the detoxification of superoxide radicals. This is a two-step process. First, superoxide dismutase (SOD) converts superoxide radicals into hydrogen peroxide, and hydrogen peroxide is subsequently converted to water and oxygen by catalase. We demonstrated in a rat model of L-N^G^-nitro-L-arginine methyl ester (L-NAME)-induced hypertension that the left ventricle can scope radical stress by upregulating mitochondria-specific SOD (SOD2), whereas the right ventricle was unable to perform such an induction [39]. This indicates two important points: first, irrespective of UCPs, mitochondria can scope oxidative stress by detoxification, and second, cell-specific differences exist that limit this protection in some tissues. On the other hand, UCPs can reduce oxidative stress by uncoupling [40]. Again, this is an evolutionary old process. Freezing stimulates the anti-oxidative capacity in *Thamnotis sirtalis* and *Rana sylvatica* independent of UCP2, whereas in *Lacerta vivipara* the upregulation of UCP is used to limit oxidative stress by re-warming [36]. Therefore, upregulation of UCP may be considered as an additional or alternative way to minimize oxidative stress in mitochondria. Nevertheless, this type of protection costs ATP generating capacity as the driving force for the ATP synthase, the proton gradient, is reduced.

A view on the role of UCPs in poikilothermic animals during re-warming is important to understand the physiological function of UCPs in other species, as the re-warming of poikilothermic animals corresponds to a physiological situation comparable with the oxidative stress that occurs in tissues during I/R. Therefore, it must be claimed that UCPs are important in reperfusion injury in mammalians, too. To sum it up, UCPs are important for metabolic flexibility, regulation of oxidative stress, fuel sensing, and ATP production.

As mentioned above, I/R in mammalians mimics evolutionary old adaptations of poikilothermic animals to re-warming. Moreover, pathomechanisms that trigger re-warming-dependent tissue damage of cold preserved organs of mammalians are similar to those mechanisms that trigger reperfusion injury. The comparison between re-warming phenomena in poikilothermic animals and reperfusion injury in endotherms and a comparison between organ re-warming and reperfusion injury within endotherms underlines similar strategies performed by cells and tissues to protect against damage. However, the situation in mammalians is more complex than those in poikilothermic animals. Mammalians express five different isoforms of UCPs, named UCP1-5 [38]. Among them, UCP1 is also known as thermogenin and expressed more or less specifically in brown adipose fat tissue, where it is responsible for thermogenesis. UCP2 and UCP3 are two structurally related isoforms of UCP1 detected in most tissues but with a tissue-dependent expression profile. UCP3 is predominantly expressed in skeletal muscles and UCP2 is predominantly expressed in hearts. UCP2 seems to be the archaic isoform from which the other isoforms originate. Finally, two other isoforms have been described in the brain, namely UCP4 and UCP5. The latter ones share less structural similarity to UCP2 than the other isoforms. They are co-expressed with UCP2 in the brain. The differential expression pattern suggests specific functions of different UCPs in tissues. Normally, in knockout animals the genetic deletion of one isoform does not induce a corresponding upregulation of another isoform; this suggests again that one isoform cannot simply replace the function of another isoform in mammalians and supports the idea that each of these isoforms has a rather specific function.

In addition to the tissue-specific distribution of the UCP isoforms, species-dependent differences add another difficulty to describing the function of UCPs in mammalians. I.e., UCP2 is barely expressed in mouse cardiomyocytes but strongly expressed in cardiomyocytes from rats, their evolutionary neighbors [41,42]. This must be kept in mind when general conclusions about the role of UCPs in I/R injury and other diseases are drawn from experiments performed in only one species.

Concerning the question of whether UCPs play a role in cardiac protection against I/R injury, we identified sixteen studies dealing with UCP2 and twenty-one studies dealing with UCP3 (search strategy: UCP2 or UCP3 and reperfusion; exclusion criteria: no peer-review journal, not published in English, review article). Concerning UCP2, 60% of all studies concluded that UCP2 protects the myocardium against reperfusion injury, while 40% of the studies came to the opposite conclusion. In contrast, nearly all studies dealing with UCP3 concluded that UCP3 triggers cardiac protection. The findings of these studies will now be discussed in detail, and we will discuss possible explanations for the different conclusions.

### 3.1. UCP2 and Cardiac Protection

First, the question of whether the expression of UCP2 is affected by I/R must be addressed. A couple of studies focused on this question. In a pig model, a two-minute ischemic period was sufficient to increase the protein expression of UCP2 in the myocardium twenty-four hours later [43]. Those hearts that showed a higher expression of UCP2 showed less oxidative stress in addition to energetic preservation. Similarly, the expression of complex IV and V was increased. Therefore, it remains unclear whether the observed protection is directly linked to UCP2 induction or the differential expression of proteins of complex IV and V. The same group published another study in which UCP2 was similarly upregulated in pigs, with stenosis developing within ten weeks [44]. Interestingly, in this case, the effect of stenosis on the expression of UCP2 was specific and not accompanied by a similar increase in UCP3. In male Wistar rats, thirteen minutes of ischemia with subsequent three hours of reperfusion were sufficient to induce the protein expression of UCP2 (but not that of UCP2 mRNA) in the left ventricle. This effect was sensitive to Losartan and Ramipril, suggesting a renin-angiotensin-dependent effect. Notably, this effect was absent in the right ventricle [45]. Using myoblasts (H9c2 cells) it was also shown that hypoxia reduces the expression of SIRT1, resulting in an induction of UCP2 expression and vice versa [46]. Activation of SIRT1 by resveratrol reduced the expression of UCP2 and vice versa [47]. Similarly, in rats a classical cardiac preconditioning protocol induced UCP2 mRNA expression within four hours and that of UCP2 protein within twenty-four hours, a process that was attenuated in the presence of the antioxidant 2-mercaptopropionyl glycin [48]. Similarly, ischemic preconditioning increases UCP2 mRNA and protein expression in a protocol of thirty minutes of ischemia and forty-five minutes of reperfusion [49]. ROS seems to induce UCP2, and the subsequent uncoupling produces a feedback inhibition, reducing ROS [50]. Seven days after myocardial infarction, UCP2 mRNA expression is still elevated with pre- and post-conditioning but not induced by I/R alone [51]. In contrast, I/R reduced the expression of UCP2 mRNA after a 45 min ischemia and subsequent 120 min of reperfusion in rats [52]. Collectively, these studies show that short periods of ischemia and reperfusion as they are used for conditioning increase UCP2 protein expression. A strict limitation of these studies is that the antibodies used to address this expression are not well-characterized, a point that is critical in the field [53]. Moreover, the corresponding mRNA either increased, is not altered, or is downregulated in different studies. Overall, the picture is not conclusive.

The second point of interest is the question of whether UCP2 expression or activity protects or damages the heart during reperfusion injury. There is no evidence that uncoupling modifies energy availability [43]. Most studies suggest that uncoupling reduces ROS, thereby protecting the heart against reperfusion injury [40,54]. Another potential mechanism by which UCP2 may protect the myocardium against reperfusion injury is by the induction of mitophagy [49]. A high expression of UCP2 may, however, damage the heart by lowering mitochondrial membrane potential and induction of apoptosis [55]. Furthermore, UCP2 may upload the mitochondria with calcium that affects heart rhythm and energy handling [56]. Collectively, the majority of studies suggest that UCP2 is required for cardiac protection. In contrast, UCP2 seems to favor fatty acid metabolism, a process that goes in concert with maladaptive hypertrophy and favors hypertrophy in general [57,58]. Overall, there may be a small window in which UCP2 activity has overall protective properties, but high uncoupling does not go alongside protection in general. Furthermore, there are strong limitations in most studies as the expression is mostly associated with I/R injury or protection but not necessarily linked; moreover, in other studies, knock-out mice were used. This species has a relatively low expression of UCP2 in myocytes [41]. Alternatively, genipin was used, which works predominately by inhibiting UCP3 in hearts [56,59]. Overall, the studies published so far are not optimal for addressing these questions.

### 3.2. UCP3 and Cardiac Protection

In contrast to a couple of studies describing the effect of ischemia or preconditioning on the expression of the UCP2 protein, there are few studies analyzing the effect of I/R and/or conditioning on the expression of UCP3. Ischemic preconditioning induces the expression of UCP2 and UCP3, but only UCP2 contributes to the loss of oxidative stress [48]. In contrast, thirty minutes of ischemia followed by two hours of reperfusion reduced the mRNA and protein expression of UCP3 [60]. Most studies addressing the question of whether UCP3 contributes to cardiac protection were performed on animals in which the initial expression of UCP3 was altered by manipulating the animals. Diabetes and insulin resistance were described as conditions under which the expression of UCP3 is altered. Diabetes increases the cardiac expression of UCP2 and UCP3 mRNA [61]. However, protein expression was increased only in the case of UCP3 [61,62]. The authors conclude that upregulation of UCP3 lowers ischemic tolerance of the heart. In contrast, UCP3 expression is decreased in insulin-resistant mice and rats [63,64,65]. Normalization of UCP3 expression improved post-ischemic recovery in these cases. Similar to diabetes, high-fat diets or maternal over-nutrition increased the cardiac expression of UCP3 [66,67,68]. The subsequent consequences for reperfusion injury are found quite heterogeneously. The upregulation of UCP3 may favor a local acidotic situation during the initial phase of reperfusion, leading to protection [66]. However, it may also increase the risk of arrhythmia and reduce ischemic tolerance [62,68]. Lastly, there are several reports considering the effect of 4-Hydroxy-2-nonenal (4-HNE), a cytotoxic product of lipid peroxidation on the expression of UCP3. HNE induces the expression of UCP3 but not that of UCP2 [59,69]. Mechanistically, HNE induces Nrf2 that, in a more general way, allows the upregulation of UCP3 [69,70]. Via this mechanism, UCP3 is considered a cardiac protective protein [71,72]. Notably, 4-HNE can also be generated by MAO-A and has been shown to improve mitochondrial Ca^2+^ overload [73]. In line with the suggestion that 4-HNE induces nuclear respiratory factor (Nrf)-2 and subsequently upregulates UCP3, such induction of UCP3 must be considered as a compensatory pathway limiting 4-HNE-induced cardiac damage. If this assumption is correct, future studies with genetic double knockouts of MAO-A and UCP3 should display a stronger I/R injury than those seen with MAO-A knockout alone. However, a direct link between MAO-A-dependent generated 4-HNE and UCP3 expression is still missing.

The protective effects of UCP3 are mechanistically linked to ROS scavenging effects but also linked to direct inhibition of the adenine nucleotide transporter (ANT) [74]. On the other hand, over-expression of ANT protects the heart against reperfusion injury and improves survival [75,76]. Therefore, it remains unclear how ANT affects I/R injury, and it remains difficult to judge the question of how UCP3 interacts with ANT and whether this is cardiac protective or induces I/R injury. Finally, UCP3 deficiency increases I/R injury and arrhythmia, although the same has also been reported under conditions of UCP3 upregulation [67,77]. The damaging effects of UCP3 have been described in diabetic hearts and after maternal over-nutrition and linked to a reduction of ischemic tolerance, but no molecular mechanisms were shown [61,68].

In conclusion, UCP3, in contrast to UCP2, seems to be regulated preferentially by metabolic alterations rather than by I/R directly (for a detailed discussion on the relationship between metabolism and UCP3 see also [78]). Furthermore, the majority of studies support the view that high UCP3 expression is protective, although this may not be the case in the presence of co-morbidities such as diabetes.

### 3.3. UCP2 and I/R Injury: General Aspects

A better understanding of the mechanisms by which uncoupling proteins may modify cardiac protection or I/R injury may be also taken from the comparison with other tissues. In the case of UCP2, this has been intensively studied for the brain, liver, and kidneys.

In the brain, UCP2 is mainly described as a neuroprotective protein based on studies with UCP2 knockout mice [79,80,81]. In these mice, UCP2 deficiency aggregates post-ischemic damage. Furthermore, studies on single nucleotide polymorphisms couple higher UCP2 expression to neuroprotection [82]. Hypoxia/reoxygenation and ischemia down-regulate the neuronal expression of UCP2 [83,84]. Nicotine, known to aggregate neuronal deficits, also downregulates UCP2 in the brain [85]. In contrast, selenium increases the expression of UCP2 and induces neuroprotective effects [86]. Mechanistically, the neuroprotective effects of UCPs were linked to ROS scavenging [87]. However, as described for the heart as well, there are a couple of studies that are in strict contrast to this view on UCP2 and neuroprotection. Occlusion of the middle cerebral arteria in diabetic rats increased the expression of UCP2 but the normalization of UCP2 expression, in this case by N-Palmitoylethanolamide-Oxazoline (PEA-OXA), went along with neuroprotection. An induction of UCP2 expression in the brain was also shown in an AMPK-dependent way or ischemia [88,89]. In both cases, the high expression of UCP2 was linked to damage. In conclusion, the normalization of otherwise decreased UCP2 expression may be neuroprotective, as well as the normalization of over-expression. The data suggest that the brain requires a constant and optimized level of UCP2. Similarly, such differential expression consequences may also explain the different outcomes in heart studies (see above).

In the liver, the expression of UCP2 was mainly investigated in studies that focus on the storage of the organ for transplantation. Studies with UCP2 knockout mice show a different picture compared to the brain and heart, as UCP2 seems not to affect hepatic reperfusion [90]. Similar conclusions were drawn with warm or cold perfusion [91,92]. Increased hepatic expression of UCP2 was found in mice fed with a high-fat diet, steatotic livers, and fat livers [93,94,95,96]. The authors suggested that a high expression of UCP2 lowers the bioavailability of ATP and thereby contributes to low outcomes. Consequently, some studies confirmed that low UCP2 expression and/or activity protects the liver against reperfusion injury [97,98,99,100]. However, other studies correlated a high expression of UCP2 with hepatic protection [101,102,103]. In conclusion, the impact of UCP2 on reperfusion injury of the liver seems to be less documented than for other tissues; however, in contrast to the heart and brain, a coupling between UCP2 and ATP has been described that was not so obvious as in other tissues [90,93,97,104]. This suggests that extramitochondrial factors affect heart-specific responses.

As in the brain, genetic deletion of UCP2 in kidney aggregates I/R injury [105,106,107]. In the absence of UCP2, the kidneys display more oxidative stress and apoptosis. Interestingly, all studies that addressed the question of whether I/R affects the expression of UCP2 found that UCP2 expression was induced. Energy deficiency as it occurs during ischemia activates AMPK that increases the expression of stanniocalcin-1 (STC1), an inductor of UCP2. This induction of UCP2 can be improved by the myokin irsin which exerts a similar UCP2-inducing effect in the lungs and pancreas [108,109]. By uncoupling, oxidative stress is reduced but ATP pools are also depleted, leading finally to a HIF1α-dependent induction of renal fibrosis [110,111,112,113,114]. Collectively, in the kidneys, the relationship between UCP2 and reperfusion injury is quite clear. An upregulation of UCP2 is required to protect the kidney against reperfusion injury. A possible explanation as to why its role in the heart is more complex may be linked to the higher energy turnover of the working heart. Overall, studies on the brain, liver, and kidneys have some limitations, such as the unclear validation of antibodies used here and the lack of control over whether genetic deletion of UCP2 causes compensatory mechanisms. Therefore, these studies suggesting tissue-specific differences in I/R injury require further validation.

Taken together, the current study revealed tissue- and species-specific differences in UCP-dependent protection. Genetic deficiency of UCP2 in rats and mice allows a better understanding of species-specific effects. The generation of UCP2 knockout rats and comparison to knockout mice may be the first step [115] (Figure 1).

## 4. Monoamine Oxidases (MAO)

### 4.1. Monoamine Oxidase Isoforms

Two different isoforms of MAO are known, namely MAO-A and MAO-B, both of which are located at the outer mitochondrial membrane. Species-dependent cardiomyocytes express different MAO isoforms: in rats, MAO-A predominates in adulthood, [116,117] while in adult mice MAO-B dominates [118,119]. Interestingly, in rat hearts, MAO-B activity also predominates up to an age of 2–3 weeks [120], most likely since MAO-B expression increases under mechanical strain as compared to the quiescent situation [121]. Human cardiomyocytes contain both MAO isoforms, but with more, albeit moderate, expression for MAO-A [122,123]. In rat hearts, MAO activity is higher in the left compared to the right ventricle [124] and females have higher plasma MAO activity than males [125] as estrogens can modulate MAO activity [126].

### 4.2. MAO Substrates

The two MAO isoforms have common substrates such as dopamine, but also specific substrates: MAO-B can metabolize 1-methyl histamine [127], produced by the histamine-N-methyltransferase [128], while MAO-A metabolizes serotonin (or 5-hydroxytryptamin, 5-HT) and catecholamines (for review, see [129]). MAO requires flavin adenine dinucleotide as a cofactor that is reduced by the reaction of, and subsequently re-oxidized by, oxygen and water, generating hydrogen peroxide [130]. MAO can also form reactive aldehydes, such as 4-hydroxynonenal, as a byproduct of catecholamine metabolism through cardiolipin peroxidation inside mitochondria in primary cardiomyocytes. Deleterious effects of 4-hydroxynonenal are physiologically prevented by the activation of mitochondrial aldehyde dehydrogenase 2 [73].

Mice deficient in both MAO-A and MAO-B demonstrate increased tissue levels of serotonin, norepinephrine, dopamine, and phenylethylamine [131], and genetic ablation of MAO-A increases the serotonin concentration in the blood and tissue of rats [132]. Similarly, a blockade of MAO by drugs indicated for other uses (e.g., antidepressants) can alter histamine levels in mice hearts [133]. In contrast, MAO-A overexpression decreases the level of norepinephrine and serotonin in the heart (for review, see [134]) (Figure 2).

### 4.3. MAO Expression

An increased expression/activity of MAO occurs during aging [135,136] and with different diseases such as arterial hypertension [137,138], pulmonary hypertension [139,140], hypertrophy [141], diabetes [142,143,144,145], myocardial infarction [146] or heart failure [147,148]. In the streptozotocin-induced diabetic rat model, particularly the MAO-B isoform is induced in aortas and hearts and contributes to the generation of reactive oxygen species [142]. While the underlying mechanisms of MAO upregulation under the above conditions are still unclear, one potential factor contributing to increased MAO expression/activity in the heart might be increased substrate availability (for review, see [134]).

An increased sympathetic tone increases plasma norepinephrine and epinephrine concentrations, and increased norepinephrine spillover as seen in chronic heart failure patients [149,150]. Serotonin concentrations are increased during different disease states (for review, see [151]) and part of the increase has been attributed to altered platelet function [152]. Histamine co-localizes with norepinephrine in neurons [153] and is enclosed in cytoplasmatic granules of mast cells, which lie adjacent to blood vessels and between cardiomyocytes [154], and mast cell degranulation might occur under stress conditions [155]. Moreover, serotonin can be formed in the mouse and human heart [156], probably by cardiomyocytes themselves [157].

In AC16 cardiomyocytes, MAO-A mRNA and protein expressions are affected by non-coding RNAs since knockdown of the non-protein coding RNA 472 (LINC00472) reduced MAO-A expression, the results being partly abolished by miR-335-3p inhibition. Thus, LINC00472 positively regulates MAOA expression via interaction with miR-335-3p [158].

### 4.4. Monoamine Oxidases and Hypertrophy

Cardiac hypertrophy is a typical early adaptive response to increased cardiac workload and mechanical stress. However, in cases of prolonged or chronic stress, this response may become maladaptive and ultimately lead to heart failure. Cardiomyocytes synthesize additional sarcomeres, leading to the thickening of the ventricular wall and increased overall cardiac mass and size. The subcellular reorganization that underlies cardiomyocyte hypertrophy was found to require functional and responsive mitochondrial dynamics (for review, see [159]).

In wild type mice, pressure overload induced by transverse aortic constriction results in increased dopamine catabolism, left ventricular hypertrophy and dilation progressing to cardiac dysfunction. In contrast, in MAO-B knockout mice with transverse aortic constriction concentric left ventricular hypertrophy and function are maintained, both at the early (weeks) and late stages (months) [160]. As outlined above, in the hearts [137] as well as in the isolated cardiomyocytes [138] of spontaneously hypertensive rats MAO activity is significantly increased even before the development of cardiac hypertrophy [138]. Increased MAO activity might represent an early event in the development of cardiac hypertrophy [137] due to its potential impact on cardiac metabolism [161] since cardiac hypertrophy normally goes along with a metabolic switch to preferential use of carbohydrates rather than fatty acids [162,163].

In rat cardiomyocytes, administration of high micromolar concentrations of serotonin or dopamine increases glucose transport through the upregulation of glucose transporters 1 and 4 at the sarcolemma; the increase in glucose import is blocked by MAO-A inhibition [161]. At similar concentrations, serotonin induces cardiomyocyte hypertrophy, again the effect being largely attenuated by MAO-A inhibition [164] (or blockade of the extracellular regulated kinase). In addition, the effects of angiotensin II on hypertrophy are attenuated by a pharmacological blockade of MAO-A in rats [165]. Since lower concentrations of serotonin induce cardiomyocyte hypertrophy independent of MAO-A through the activation of the 5-HT(A2) receptor [166,167], genetic deletion of MAO-A also increases load-dependent ventricular hypertrophy [132] (for review, see [117]). Thus, both an increased or decreased MAO expression/activity can contribute to hypertrophic effects, depending on substrate availability.

### 4.5. Pulmonary Hypertension

MAOs have also been proposed to play an important role in pulmonary hypertension [168]. In rats, pulmonary hypertension secondary to monocrotaline injection [139], sugen5416/hypoxia, or pulmonary artery banding [140] upregulates MAO-A expression in the pulmonary vasculature and the failing right ventricle. Clorgyline treatment reduced the right ventricular afterload and pulmonary vascular remodeling in sugen/hypoxia rats through reduced pulmonary vascular proliferation and oxidative stress, resulting in improved right ventricular stiffness and relaxation and reversed right ventricular hypertrophy. In rats with pulmonary artery banding, clorgyline had no direct effect on the right ventricle [140]. In contrast, recent unpublished data demonstrate less myocardial structural or functional changes secondary to the pulmonary artery banding in cardiomyocyte-specific MAO B knockout mice in the right ventricle.

### 4.6. Monoamine Oxidases and Ischemia/Reperfusion (I/R) Injury (for Review Also See [134,169])

Under stress conditions such as I/R, the autonomic nervous system is activated, releasing neurotransmitters that are metabolized by MAOs, thereby directly influencing heart function [169]. Besides norepinephrine, serotonin and histamine also play important roles in I/R injury. Serotonin accumulates in the heart during ischemia [170] and is degraded after reperfusion depending on MAO-A activity after uptake into cells [171]. Mast cells become activated during stress conditions and release histamine [155]; histamine release from the heart is increased during I/R [127]. While mast cell activation thus increases substrate availability for MAO-B, MAO-B inhibition prevents mast cell degranulation in diabetic mice hearts [144], implying a vicious cycle of mast cell and MAO-B activation.

During 30 min ischemia, hydroxyl radical production increases 2-fold with a further increase upon 60 min reperfusion in isolated rat hearts, both of which can be decreased by pargyline administration. The decrease in ROS formation following MAO inhibition is associated with reduced cardiomyocyte injury following I/R [172]. Similarly, cardiomyocyte-specific MAO-B knockout reduces infarct size following I/R in isolated mice hearts [173].

In vivo, the inhibition of MAO-A largely reduces myocardial ultrastructural damage induced by 30 min ischemia and 60 min reperfusion in the rat heart, associated with the prevention of postischemic oxidative stress, neutrophil accumulation, and mitochondrial-dependent cell death [174]. Infarct size and cardiomyocyte apoptosis are also significantly decreased in MAO-A-deficient animals following 30 min ischemia and 180 min reperfusion, the protection being accompanied by sphingosine kinase 1 inhibition and less ceramide accumulation [175].

Cardioprotection can be achieved also by mechanical intervention such as ischemic preconditioning [176]. In both male and female rat hearts, ischemic preconditioning improves functional recovery following I/R, which is further enhanced in the presence of MAO inhibition by either clorgyline or pargyline. However, infarct size is similar among all preconditioned groups, regardless of the presence of MAO inhibitors, indicating that acute inhibition of MAOs potentiates the preconditioning-induced postischemic functional recovery without having any further effect on infarct size beyond that achieved by ischemic preconditioning [177].

### 4.7. Monoamine Oxidases and Left Ventricular Remodeling/Heart Failure

Rasagiline mesylate (N-propargyl-1 (R)-aminoindan) (RG), a selective, potent irreversible inhibitor of monoamine oxidase-B, administered for 28 days (2 mg/kg) starting 24 h after myocardial infarction, preserves left ventricular geometry and function. Treatment with rasagiline prevents tissue fibrosis and attenuates cardiomyocyte apoptosis in the border zone of the infarct associated with a markedly-decreased malondialdehyde level in the border zone, indicating a reduction in tissue oxidative stress [178]. Additionally, MAO-A is an important source of oxidative stress in the heart and MAO-A-derived reactive oxygen species contribute to dilated cardiomyopathy [145]. In mice, left ventricular function following four weeks of coronary artery occlusion improves by pharmacological or genetic inhibition of MAO-A. Both interventions protect the mice from 4-hydroxynonenal accumulation and mitochondrial calcium overload, thus mitigating ventricular dysfunction [73].

Furthermore, it has recently been suggested that upregulation of MAO-A during heart failure will accelerate intracellular catecholamine degradation, thereby inhibiting a direct stimulation of β-adrenergic receptors at the sarcoplasmic reticulum; this interaction is closely linked to phospholamban phosphorylation and calcium filling of the sarcoplasmic reticulum [179,180].

The importance of MAO for left ventricular remodeling and heart failure development can be shown in mice with chronic overexpression of MAO-A. Here, reactive oxygen species [136] and 4-hydroxynonenal concentrations increase, followed by mitochondrial dysfunction [181], cardiomyocyte hypertrophy, reduced left ventricular function and increased cardiac fibrosis [73], as well as increased cardiac inflammation [182] (for review, see [134,183]). When transgenic animals are treated with the antioxidant N-acetyl cysteine part of the above effects can be rescued [136,184].

Moreover, heart failure induced by doxorubicin is affected by MAO inhibition, which prevents both the severe oxidative stress induced by doxorubicin as well as chamber dilation and cardiac dysfunction in doxorubicin-treated mice in vivo [185].

## 5. Conclusions

The current data in the literature suggest that I/R will preferentially upregulate UCP2 in cardiac tissue but not UCP3. Studies addressing the consequences of such an induction are currently not conclusive because the precise function of UCP2 in cardiac tissue is not well understood and is complicated further by tissue- and species-specific aspects. In general, UCP2 may reduce oxidative stress by mild uncoupling and both UCP2 and UCP3 affect substrate utilization in cardiac tissue, thereby modifying post-ischemic remodeling. MAOs are important for the physiological regulation of substrate concentrations (such as in the case of serotonin and catecholamines). Upon increased expression and or activity of MAOs, however, the increased production of reactive oxygen species and reactive aldehydes contributes to cardiac alterations such as hypertrophy, inflammation, irreversible cardiomyocyte injury, and failure.

## Figures and Tables

**Figure 1 ijms-24-06459-f001:**
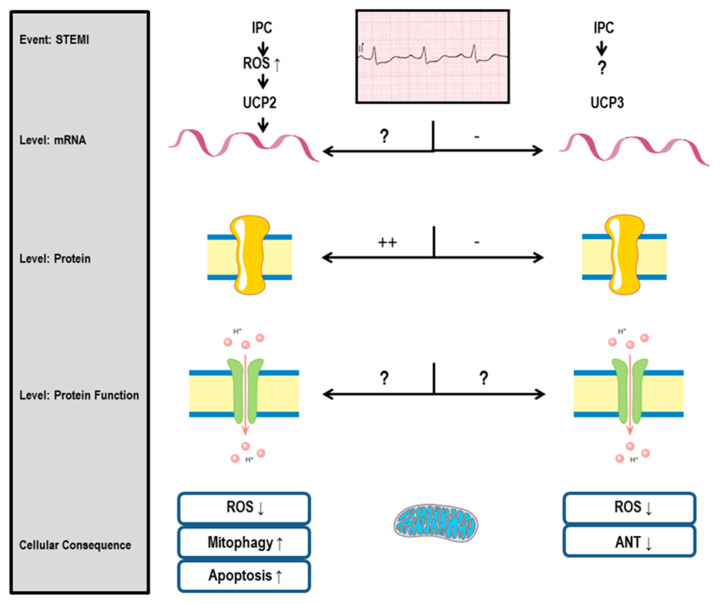
Myocardial Infarction (STEMI) strongly induces uncoupling protein (UCP) 2 expression (level protein). Short periods of ischemia, as performed by ischemic preconditioning (IPC), can also induce UCP2 mRNA expression (level mRNA). Whether this increases the protonophoric function of UCP2 has not been analyzed (levels protein function). The cellular consequences are linked to altered mitochondrial function and include reduced reactive oxygen species (ROS) production and increased mitophagy, but also increased apoptosis. In contrast, STEMI seems to reduce the mRNA and protein expression of UCP3. Decreased expression of UCP3 lowers protection via ROS reduction and adenine nucleotide transporter (ANT) inhibition, which was linked to UCP3 but not UCP2 (see details in text).

**Figure 2 ijms-24-06459-f002:**
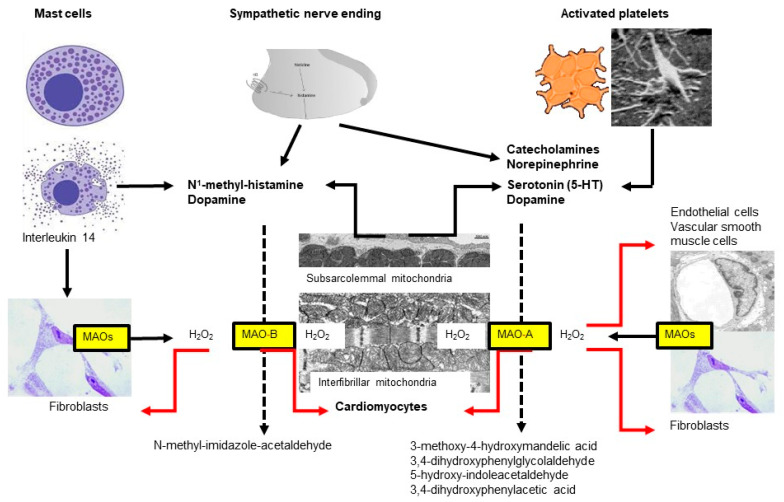
Two subtypes of monoamine oxidases (MAO)—named A and B—are located at the outer mitochondrial membrane, which differ in their substrate specificity. Almost all cell types express MAOs but the respective subtype might differ between species, organs and age. In the heart, MAOs are expressed in cardiomyocytes, fibroblasts, vascular smooth muscle and endothelial cells. In the heart, MAO substrates are derived from different sources including mast cells, sympathetic nerves, platelets and cardiomyocytes.

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
