# Peer review of "Importance of Mitochondria in Cardiac Pathologies: Focus on Uncoupling Proteins and Monoamine Oxidases"

_ijms, 2023, doi:10.3390/ijms24076459_

Round 1
Reviewer 1 Report
The review article by Schulz and Schlüter is well written and covers most of the current knowledge in the field of reactive oxygen species and cardiac pathologies.
I would suggest adding a short paragraph about the contribution of ROS derived from non-mitochondrial sources including xanthine oxidase, uncoupled nitric oxide synthase, NADPH oxidase etc., in the pathophysiology of the heart.
Author Response
Responses to Reviewer#1:
The review article by Schulz and Schlüter is well written and covers most of the current knowledge in the field of reactive oxygen species and cardiac pathologies.
I would suggest adding a short paragraph about the contribution of ROS derived from non-mitochondrial sources including xanthine oxidase, uncoupled nitric oxide synthase, NADPH oxidase etc., in the pathophysiology of the heart.
Response:
We would like to thank the reviewer for her/his positive judgement on our review article and the suggestion to add a paragraph on non-mitochondrial sources of ROS. However, as pointed out in the title of the review article we would like to concentrate on mitochondrial ROS and have chosen two mitochondrial proteins, one located at the inner mitochondrial membrane (UCP) and one located at the outer mitochondrial membrane (MAO), which play a role in the overall balance of ROS formation by mitochondria. Other mitochondrial proteins involved in ROS formation have been discussed by us more recently (overall importance [1, 2], connexin 43 [3], STAT [4], respiratory chain [5], p66shc [6], NOX4 [7]) as well as the antioxidative defense ([8]) and were therefore not part of the current review article. We hope for your understanding.
We reworded the paragraph; it now reads:
Intracellular ROS are derived from various enzymes (xanthine oxidase, uncoupled nitric oxide synthase, sodium-potassium ATPase, and nicotinamide adenine dinucleotide phosphate (NADPH) oxidase). Also, in mitochondria apart from the electron transport chain a number or proteins have been identified which contribute to ROS formation, namely connexin 43 [3], signal transducer and activator of transcription [4], or p66shc [6] as previously discussed by us (for review, also see [9]). We now concentrate on two other mitochondrial proteins, i.e. uncoupling proteins (UCP, depending membrane on the mitochondrial membrane potential) and monoamine oxidase (MAO).
- Andreadou, I.; Schulz, R.; Papapetropoulos, A.; Turan, B.; Ytrehus, K.; Ferdinandy, P.; Daiber, A.; Di Lisa, F., The role of mitochondrial reactive oxygen species, NO and H2 S in ischaemia/reperfusion injury and cardioprotection. J Cell Mol Med 2020, 24, (12), 6510-6522.
- Daiber, A.; Steven, S.; Euler, G.; Schulz, R., Vascular and Cardiac Oxidative Stress and Inflammation as Targets for Cardioprotection. Curr Pharm Des 2021, 27, (18), 2112-2130.
- Boengler, K.; Leybaert, L.; Ruiz-Meana, M.; Schulz, R., Connexin 43 in Mitochondria: What Do We Really Know About Its Function? Front Physiol 2022, 13, 928934.
- Comita, S.; Femmino, S.; Thairi, C.; Alloatti, G.; Boengler, K.; Pagliaro, P.; Penna, C., Regulation of STAT3 and its role in cardioprotection by conditioning: focus on non-genomic roles targeting mitochondrial function. Basic Res Cardiol 2021, 116, (1), 56.
- Szibor, M.; Schreckenberg, R.; Gizatullina, Z.; Dufour, E.; Wiesnet, M.; Dhandapani, P. K.; Debska-Vielhaber, G.; Heidler, J.; Wittig, I.; Nyman, T. A.; Gartner, U.; Hall, A. R.; Pell, V.; Viscomi, C.; Krieg, T.; Murphy, M. P.; Braun, T.; Gellerich, F. N.; Schluter, K. D.; Jacobs, H. T., Respiratory chain signalling is essential for adaptive remodelling following cardiac ischaemia. J Cell Mol Med 2020, 24, (6), 3534-3548.
- Boengler, K.; Bornbaum, J.; Schluter, K. D.; Schulz, R., P66shc and its role in ischemic cardiovascular diseases. Basic Res Cardiol 2019, 114, (4), 29.
- Hirschhauser, C.; Bornbaum, J.; Reis, A.; Bohme, S.; Kaludercic, N.; Menabo, R.; Di Lisa, F.; Boengler, K.; Shah, A. M.; Schulz, R.; Schmidt, H. H., NOX4 in Mitochondria: Yeast Two-Hybrid-Based Interaction with Complex I Without Relevance for Basal Reactive Oxygen Species? Antioxid Redox Signal 2015, 23, (14), 1106-12.
- Andreadou, I.; Efentakis, P.; Frenis, K.; Daiber, A.; Schulz, R., Thiol-based redox-active proteins as cardioprotective therapeutic agents in cardiovascular diseases. Basic Res Cardiol 2021, 116, (1), 44.
- Schluter, K. D.; Kutsche, H. S.; Hirschhauser, C.; Schreckenberg, R.; Schulz, R., Review on Chamber-Specific Differences in Right and Left Heart Reactive Oxygen Species Handling. Front Physiol 2018, 9, 1799.

Reviewer 2 Report
The review "Importance of mitochondria in cardiac pathologies: Focus on uncoupling proteins and monoamine oxidases" by Rainer Schulz and Klaus-Dieter Schlüter focuses on the complex literature trying to understand the physiological importance of reactive oxygen species (ROS) as signaling molecules and the role of UCPs and MAO in this process. Both authors are internationally recognized researchers in the field of myocardial ischemia/reperfusion injury (IRI), investigating the importance of mitochondria and mitochondria-derived ROS in IRI.
The manuscript refers only to published data. The review is highly topical and can contribute to many controversial issues in the reviewed field. Unfortunately, at a current stage many discussed issues lack a critical evaluation, new reliable data and well-founded hypotheses are not presented and not discussed. The review can benefit a lot if it is written much more consistently. Many unnecessary repetitions can be avoided.
Following major issues can be significantly improved.
1. The title "Importance of mitochondria in cardiac pathologies ..." suggests that only topics related to the heart will be discussed. Discussions in various places devoted to the brain, kidney, liver, and thermogenesis in BAT defocus the review.
2. The particular interest in two different issues that are not really connected - uncoupling proteins and monoamine oxidases - is not clearly explained.
3. In the abstract the authors write “Apart from the electron transport chain, uncoupling protein (UCP) and monoamine oxidases (MAO) have been proposed to contribute to mitochondrial ROS formation”. Indeed, monoamine oxidase is one of the important sources of ROS in the mitochondria. In contrast UCPs are thought to decrease the mitochondrial ROS.
4. It would be also helpful if the abstract included more specific information about the topics covered in the review.
5. Line 79-165.
A long introduction about “Uncoupling proteins” starting from the line 79 doesn´t make sense in this review and is inaccurate. So, a meaningful protonophoric function (uncoupling) is only accepted by the community for UCP1, because of its strong upregulation under cold-acclimated conditions and NOT because proteins higher proton turnover number as stated by the authors (protonophoric capacity, line 137). Proton turnover number (protonophoric activity) was shown to be roughly the same for different recombinant uncoupling proteins (UCP1-UCP3).
“Mild uncoupling” is relevant only for the homologous proteins UCP2-UCP4, because their low expression and consequently inability to generate heat.
General UCP description does not reflect a lot of controversy in this field and the reasons for this.
I think that if the discussion will focus on UCP3 and UCP2, with a brief description of the subfamily to which they belong, the introduction would be much more consistent.
6. When one thinks of the heart, one thinks of UCP3 because its expression in the heart has been shown and accepted by most groups for years. The recent literature even connects the presence of UCP3 in heart with its main metabolism - fatty acid oxidation (10.3389/fphys.2019.00470). It would be interesting, if authors as known heart specialists will present their opinion to this new development that is completely ignored at the present manuscript.
7. The expression and role of UCP2 in heart is very much disputed. The studies investigating the expression of UCP2 in heart should be critically evaluated for the correct evidence (Western Blots), such as use of evaluated antibodies and appropriate controls. It should be mentioned that gene expression cannot be equated with protein expression because the lifetime of UCP2 is very short. Authors should discuss whether the appearance of UCP2 in heart may be connected to the situations, which require the increase of glycolysis, such as development of cancer or increase in stem or inflammatory cells. It would be very valuable, if authors could connect their text as cited below with expression of UCP3 or UCP2.
Line 70-73: Increases in glucose uptake and glycolysis in rodent hearts are mandatory for protecting the heart during low-flow ischemia [30]. Increased fatty acid uptake and incomplete fatty acid metabolism during ischemia aggravate I/R injury through the build-up of long-chain acylcarnitines within the mitochondria, resulting in an increase of mitochondrial ROS production
8. In the context of the possible involvement of UCP2 and UCP3 in metabolism, it would be interesting to discuss the role of UCP2 in the failing heart as a possible transporter of C4 metabolites (10.1073/pnas.1317400111).
9. Line 259-309. The papers investigating the role of UCP2 in brain, liver, and kidney should be carefully and critically evaluated. The studies using evaluated antibodies and positive/negative controls showed no UCP2 in these organs (specific cells like neurons, hepatocytes, etc.). Surely, microglia express UCP2 and it is also possible that the invasion of the immunocompetent cells (expressing UCP2 especially by activation) can lead to the visible bands in Western Blots.
In the KO studies usually don´t consider and don´t evaluate the simultaneous down-regulation of other potentially relevant genes. This issue should be discussed.
10. It may be interesting to connect the recently described increase of local 4-HNE production in heart due to MAO-A activation with the increase of HNE-mediated protonophoric activity of uncoupling proteins in the presence of fatty acids.
11. The manuscript will benefit from the additional language and typos check. Below few examples.
Line 28 “UCP, depending n the mitochondrial membrane potential”
Line 30 “importance of UCP and MAO with cardiac injury “à “importance of UCP and MAO in (mechanism of?) cardiac injury “
12. Several sentences (just an example) should be reworded for clarity:
“Cardiac mitochondria are dynamic organelles and organize into differentiated populations and decreasing mitochondrial fission (or increasing mitochondrial fusion) reduce I/R injury”

Author Response
Responses to Reviewer#2:
The review "Importance of mitochondria in cardiac pathologies: Focus on uncoupling proteins and monoamine oxidases" by Rainer Schulz and Klaus-Dieter Schlüter focuses on the complex literature trying to understand the physiological importance of reactive oxygen species (ROS) as signaling molecules and the role of UCPs and MAO in this process. Both authors are internationally recognized researchers in the field of myocardial ischemia/reperfusion injury (IRI), investigating the importance of mitochondria and mitochondria-derived ROS in IRI.
The manuscript refers only to published data. The review is highly topical and can contribute to many controversial issues in the reviewed field. Unfortunately, at a current stage many discussed issues lack a critical evaluation, new reliable data and well-founded hypotheses are not presented and not discussed. The review can benefit a lot if it is written much more consistently. Many unnecessary repetitions can be avoided.
We would like to thank the reviewer for her/his positive judgement on our review article and the suggestions which we have tried to incorporate in the revised version of the review article.
Following major issues can be significantly improved.
- The title "Importance of mitochondria in cardiac pathologies ..." suggests that only topics related to the heart will be discussed. Discussions in various places devoted to the brain, kidney, liver, and thermogenesis in BAT defocus the review.
Response: At first glance discussions that compare aspects of UCP2 physiology with those linked to the role of UCP2 in ischemia/reperfusion injury of the heart may be considered as out of focus of the review topic. However, in the context of cardiac damage it is important to understand whether the observed contribution of UCP2 is tissue specific or not. This is in particular important for the UCP2 part of our review article because here the comparison between brain, liver, and kidney suggests tissue specific differences. This is in clear contrast to MAOs which shows less tissue-specific variations. Therefore, we clearly state in our manuscript: “A better understanding about mechanisms by which uncoupling proteins may modify cardiac protection or I/R injury may be also taken from the comparison with other tissues. In case of UCP2 this has been intensively studied for the brain, liver, and kidney.”
- The particular interest in two different issues that are not really connected - uncoupling proteins and monoamine oxidases - is not clearly explained.
Response: The title of our review claims the importance of mitochondria in cardiac pathologies. Mitochondria can modulate the excess of ROS by production and scavenging of ROS. This is not entirely linked to the inner mitochondrial membrane but also linked to enzymes located in the outer mitochondrial membrane. Here, we focus on UCPs as a member of the inner mitochondrial membrane that modifies ROS formation depending on the membrane potential (see Introduction) and on MAO located at the outer mitochondrial membrane that produces ROS depending on substrate availability. As clearly stated in the introduction these two proteins play a role on mitochondria-dependent ROS formation. On the other hand the term ‘play a role’ also indicates that there are other possible sources of mitochondria-dependent ROS formation and of course proteins of the mitochondrial intermembrane space and the matrix that contribute to the overall balance of ROS formation by mitochondria. Therefore, the authors are aware of the fact that they describe not the role of mitochondria for ROS formation in total but focus on two proteins in both the intra- und outer-mitochondrial membrane that require further attention. Other mitochondrial proteins involved in ROS formation have been discussed by us more recently (overall importance [1, 2], connexin 43 [3], STAT [4], respiratory chain [5], p66shc [6], NOX4 [7]) as well as the antioxidative defense ([8]) and were therefore not part of the current review article. We hope for your understanding.
- In the abstract the authors write “Apart from the electron transport chain, uncoupling protein (UCP) and monoamine oxidases (MAO) have been proposed to contribute to mitochondrial ROS formation”. Indeed, monoamine oxidase is one of the important sources of ROS in the mitochondria. In contrast UCPs are thought to decrease the mitochondrial ROS.
Response: We totally agree to the comment of the reviewer that UCP will preferentially reduce ROS rather than increase ROS production. We considered the term ‘contribute’ in both directions: Increase or decrease of oxidative burst. However, we changes the sentence to be more precise: Please read now: “Apart from the electron transport chain, uncoupling protein (UCP) and monoamine oxidases (MAO) have been proposed to modify mitochondrial ROS formation”
- It would be also helpful if the abstract included more specific information about the topics covered in the review.
Response: We agree to the reviewers comment that the abstract can be more informative if key conclusions from our review article are given. We have changed the review accordingly.
- Line 79-165. A long introduction about “Uncoupling proteins” starting from the line 79 doesn´t make sense in this review and is inaccurate. So, a meaningful protonophoric function (uncoupling) is only accepted by the community for UCP1, because of its strong upregulation under cold-acclimated conditions and NOT because proteins higher proton turnover number as stated by the authors (protonophoric capacity, line 137). Proton turnover number (protonophoric activity) was shown to be roughly the same for different recombinant uncoupling proteins (UCP1-UCP3). “Mild uncoupling” is relevant only for the homologous proteins UCP2-UCP4, because their low expression and consequently inability to generate heat. General UCP description does not reflect a lot of controversy in this field and the reasons for this. I think that if the discussion will focus on UCP3 and UCP2, with a brief description of the subfamily to which they belong, the introduction would be much more consistent.
Response: We have rephrased this part and removed the discussion about ‘mild’ uncoupling. We agree that the amount of uncoupling depends on the relative protein expression level. In the first part we acknowledge that uncoupling and heat production are evolutionary old pathways which are linked to substrate metabolism. We consider this as essential for the following discussions about the role of UCPs in I/R. In the second part we introduce the effect of substrate utilization on expression of uncoupling proteins that lead to the subsequent findings in most studies with I/R that focus on a potential role of UCPS. Therefore, we consider this part as essential for our review. In the third part of this chapter we compare the role of UCPs with those of other strategies of ROS defense in mitochondria and address the point that both strategies are cell- and tissue-specific. Again, we consider this essential to understand the studies cited thereafter on I/R. In the next part we explain why the evolutionary old pathway is important for I/R in mammalians. Again, it is important for the subsequent discussion to understand that re-warming and I/R are producing similar challenges for cells. In the next part we introduce the more relevant isoforms (UCP2, 3) for mammalian hearts. UCP1, 4, and 5 are only mentioned and explained that they are NOT expressed in cardiac tissue. This is not a big introduction of other isoforms, there existence is simply acknowledged. Thereafter it is mentioned that species-dependent differences are taken into account. The last paragraph then describes how the literature was searched. We consider the remaining parts of the discussion as essential for the understanding of the subsequent cited studies.
- When one thinks of the heart, one thinks of UCP3 because its expression in the heart has been shown and accepted by most groups for years. The recent literature even connects the presence of UCP3 in heart with its main metabolism - fatty acid oxidation (10.3389/fphys.2019.00470). It would be interesting, if authors as known heart specialists will present their opinion to this new development that is completely ignored at the present manuscript.
Response: There is no doubt that UCP3 is expressed in the heart. We have clearly stated this in our review although in accordance to [9] the expression of UCP3 in the skeletal muscle exceeds that of UCP3 in the heart. The conclusion of chapter 3.2. clearly states your suggestion: “…UCP3 in contrast to UCP2 seems to be regulated preferentially by metabolic alterations rather than by I/R directly.” We now refer to [9] allowing the reader to get more information about metabolism and UCP3.
- The expression and role of UCP2 in heart is very much disputed. The studies investigating the expression of UCP2 in heart should be critically evaluated for the correct evidence (Western Blots), such as use of evaluated antibodies and appropriate controls. It should be mentioned that gene expression cannot be equated with protein expression because the lifetime of UCP2 is very short. Authors should discuss whether the appearance of UCP2 in heart may be connected to the situations, which require the increase of glycolysis, such as development of cancer or increase in stem or inflammatory cells. It would be very valuable, if authors could connect their text as cited below with expression of UCP3 or UCP2.
Line 70-73: Increases in glucose uptake and glycolysis in rodent hearts are mandatory for protecting the heart during low-flow ischemia [30]. Increased fatty acid uptake and incomplete fatty acid metabolism during ischemia aggravate I/R injury through the build-up of long-chain acylcarnitines within the mitochondria, resulting in an increase of mitochondrial ROS production
Response: The reviewer raises an important point about the expression of UCP2 in the heart and in particular in cardiomyocytes. Please note that we have clearly stated this in our review article: “A strict limitation of these studies is that the antibodies used to address this expression are not well characterized a point that is critical in the field [10]. Moreover, the corresponding mRNA either increased, was not altered or down-regulated in different studies. Overall the picture is not conclusive.“
Furthermore, our own study showed UCP2 protein expression in rat cardiomyocytes with an antibody validated with tissue samples from UCP2-/- mice and rat cardiomyocytes in which mRNA expression was targeted by siRNA directed against UCP2. The experiments with siRNA clarify that protein expression can be affected by targeting mRNA expression (ref. 40). Due to the rather low expression of UCP2 in cardiomyocytes from mice, studies with genetic deletion of UCP2 in rats would be mandatory to clarify the questions whether UCP2 is required for cardiac protection or not. Please note that in a classical in vitro approach (Langendorff technique) the contribution of neither stem cells nor that of immune cells can be addressed. This requires a further study investigating I/R injury in vivo. However, such data are currently not available and therefore they cannot be topic of our review.
- In the context of the possible involvement of UCP2 and UCP3 in metabolism, it would be interesting to discuss the role of UCP2 in the failing heart as a possible transporter of C4 metabolites (10.1073/pnas.1317400111).
Response: Concerning heart failure in general, the heart shows a metabolic switch from the preferential use of fatty acids to the use of glucose. A role for UCP2 in this metabolic adaptation has been suggested by several authors. It has been described before that reduced UCP2 expression (both protein and mRNA) goes along with increased usage of glucose. In contrast, 10.1073/pnas.1317400111 describes a complete different mechanism by which increased UCP2 activity favors non-oxidative use of glucose in tumor cells and increases glutaminolysis. However, there are currently no data available whether a similar mechanism occurs in the heart or non-tumor cells. Own non published data showed cardiotoxic effects of glutamine if it replaced glucose. Overall, we consider the function of UCP2 as a C4 transporter in the heart as not proven to date. Furthermore, such a function would be directed against other findings. As the review is aimed to summarize the published understanding for the heart we did not discuss this interesting finding from tumor cells here.
- Line 259-309. The papers investigating the role of UCP2 in brain, liver, and kidney should be carefully and critically evaluated. The studies using evaluated antibodies and positive/negative controls showed no UCP2 in these organs (specific cells like neurons, hepatocytes, etc.). Surely, microglia express UCP2 and it is also possible that the invasion of the immunocompetent cells (expressing UCP2 especially by activation) can lead to the visible bands in Western Blots. In the KO studies usually don´t consider and don´t evaluate the simultaneous downregulation of other potentially relevant genes. This issue should be discussed.
Response: As explained above these studies should allow the reader to compare tissue-specific and tissue-unspecific effects described for the heart. Concerning the brain, liver, and kidney the majority of studies addressing the role of UCP2 in I/R injury use knockout mice and describe differences between knockout mice and wildtype mice. We agree that a limitation of such an analysis is the unknown question of compensatory gene regulation. However, in all three tissues genetic deletion leads to alterations in I/R injury that at least in part differed from that of the heart. As described in the review, energy demand and mitochondrial function may significantly differ between all these tissues. We have now included a short paragraph addressing these limitations.
- It may be interesting to connect the recently described increase of local 4-HNE production in heart due to MAO-A activation with the increase of HNE-mediated protonophoric activity of uncoupling proteins in the presence of fatty acids.
Response: We thank the reviewer for this important advise. 4-Hydroxynonenal (4-HNE) has been linked to the induction of UCP3 not UCP2. As described in our review, the data published so far suggest that UCP3 protects the heart against I/R injury. If this assumption is correct, 4-HNE should not damage the heart. Please read in chapter 3.2.: “Of note, 4-HNE can also be generated by MAO-A and has been shown to improve mitochondrial Ca2+ overload [11]. In line with the suggestion that 4-HNE induces nuclear respiratory factor (Nrf)-2 and subsequently upregulates UCP3 such an induction of UCP3 must be considered as a compensatory pathway limiting 4-HNE-induced cardiac damage. If this assumption is correct, future studies with genetic double knockouts of MAO-A and UCP3 should display a stronger I/R injury than the one seen with MAO-A knockout alone. However, a direct link between MAO-A-dependent generated 4-HNE and UCP3 expression is still missing.”
- The manuscript will benefit from the additional language and typos check. Below few examples.
Response: done
Line 28 “UCP, depending n the mitochondrial membrane potential”
Line 30 “importance of UCP and MAO with cardiac injury “à “importance of UCP and MAO in (mechanism of?) cardiac injury “
- Several sentences (just an example) should be reworded for clarity:
“Cardiac mitochondria are dynamic organelles and organize into differentiated populations and decreasing mitochondrial fission (or increasing mitochondrial fusion) reduce I/R injury”
Response: done
- Andreadou, I.; Schulz, R.; Papapetropoulos, A.; Turan, B.; Ytrehus, K.; Ferdinandy, P.; Daiber, A.; Di Lisa, F., The role of mitochondrial reactive oxygen species, NO and H2 S in ischaemia/reperfusion injury and cardioprotection. J Cell Mol Med 2020, 24, (12), 6510-6522.
- Daiber, A.; Steven, S.; Euler, G.; Schulz, R., Vascular and Cardiac Oxidative Stress and Inflammation as Targets for Cardioprotection. Curr Pharm Des 2021, 27, (18), 2112-2130.
- Boengler, K.; Leybaert, L.; Ruiz-Meana, M.; Schulz, R., Connexin 43 in Mitochondria: What Do We Really Know About Its Function? Front Physiol 2022, 13, 928934.
- Comita, S.; Femmino, S.; Thairi, C.; Alloatti, G.; Boengler, K.; Pagliaro, P.; Penna, C., Regulation of STAT3 and its role in cardioprotection by conditioning: focus on non-genomic roles targeting mitochondrial function. Basic Res Cardiol 2021, 116, (1), 56.
- Szibor, M.; Schreckenberg, R.; Gizatullina, Z.; Dufour, E.; Wiesnet, M.; Dhandapani, P. K.; Debska-Vielhaber, G.; Heidler, J.; Wittig, I.; Nyman, T. A.; Gartner, U.; Hall, A. R.; Pell, V.; Viscomi, C.; Krieg, T.; Murphy, M. P.; Braun, T.; Gellerich, F. N.; Schluter, K. D.; Jacobs, H. T., Respiratory chain signalling is essential for adaptive remodelling following cardiac ischaemia. J Cell Mol Med 2020, 24, (6), 3534-3548.
- Boengler, K.; Bornbaum, J.; Schluter, K. D.; Schulz, R., P66shc and its role in ischemic cardiovascular diseases. Basic Res Cardiol 2019, 114, (4), 29.
- Hirschhauser, C.; Bornbaum, J.; Reis, A.; Bohme, S.; Kaludercic, N.; Menabo, R.; Di Lisa, F.; Boengler, K.; Shah, A. M.; Schulz, R.; Schmidt, H. H., NOX4 in Mitochondria: Yeast Two-Hybrid-Based Interaction with Complex I Without Relevance for Basal Reactive Oxygen Species? Antioxid Redox Signal 2015, 23, (14), 1106-12.
- Andreadou, I.; Efentakis, P.; Frenis, K.; Daiber, A.; Schulz, R., Thiol-based redox-active proteins as cardioprotective therapeutic agents in cardiovascular diseases. Basic Res Cardiol 2021, 116, (1), 44.
- Pohl, E. E.; Rupprecht, A.; Macher, G.; Hilse, K. E., Important Trends in UCP3 Investigation. Front Physiol 2019, 10, 470.
- Rupprecht, A.; Brauer, A. U.; Smorodchenko, A.; Goyn, J.; Hilse, K. E.; Shabalina, I. G.; Infante-Duarte, C.; Pohl, E. E., Quantification of uncoupling protein 2 reveals its main expression in immune cells and selective up-regulation during T-cell proliferation. PLoS One 2012, 7, (8), e41406.
- Santin, Y.; Fazal, L.; Sainte-Marie, Y.; Sicard, P.; Maggiorani, D.; Tortosa, F.; Yucel, Y. Y.; Teyssedre, L.; Rouquette, J.; Marcellin, M.; Vindis, C.; Shih, J. C.; Lairez, O.; Burlet-Schiltz, O.; Parini, A.; Lezoualc'h, F.; Mialet-Perez, J., Mitochondrial 4-HNE derived from MAO-A promotes mitoCa(2+) overload in chronic postischemic cardiac remodeling. Cell Death Differ 2020, 27, (6), 1907-1923.

Reviewer 3 Report
This paper updates the reader about the recent findings in the role of UCP and MAO in cardiac injury. It updates to the most recent findings. The novelty is in the content analysis and in the paper approach in matching information related to UCP and MAO and cardiac injuries. it opens to some possible alternative markers to evaluate.
Please add a conclusion at the end of the article to summarize the founding.
In the present format, it is difficult to have a complete full view of the issue you are tackling.
Author Response
Response to reviewer#3:
This paper updates the reader about the recent findings in the role of UCP and MAO in cardiac injury. It updates to the most recent findings. The novelty is in the content analysis and in the paper approach in matching information related to UCP and MAO and cardiac injuries. it opens to some possible alternative markers to evaluate.
Please add a conclusion at the end of the article to summarize the founding. In the present format, it is difficult to have a complete full view of the issue you are tackling. The paragraph now reads:
We would like to thank the reviewer for her/his positive judgement on our review article and the suggestion to add a conclusion paragraph. The paragraph now reads:
“Conclusion: The current data in the literature suggest that I/R will preferentially upregulate UCP2 in cardiac tissue but not UCP3. Studies addressing the consequences of such an induction are currently not conclusive because the precise function of UCP2 in cardiac tissue is not well understood and tissue-specific as well as species-specific aspects complicate the situation. In general, UCP2 may reduce oxidative stress by mild uncoupling and both UCP2 and UCP3 affect substrate utilization in cardiac tissue thereby modifying post-ischemic remodeling. Taken together, MAOs are important for the physiological regulation of substrate concentrations (such as in the case of serotonin and catecholamines). Upon increased expression and or activity of MAOs, however, the increased production of reactive oxygen species and reactive aldehydes contribute to cardiac alterations such as hypertrophy, inflammation, irreversible cardiomyocyte injury and failure.”
